# Laplacian pyramid-based complex neural network learning for fast MR imaging

**Haoyun Liang**[*1]
**Yu Gong**[*1]
**Hoel Kervadec**[2]
**Cheng Li**[1]
**Jing Yuan**[3]
**Xin Liu**[1]
**Hairong Zheng**[1]
**Shanshan Wang**[†1]                    SS.WANG@SIAT.AC.CN; SOPHIASSWANG@HOTMAIL.COM
[1] *Paul C. Lauterbur Research Center for Biomedical Imaging, Shenzhen Institutes of Advanced Technology, Chinese Academy of Sciences, Shenzhen, Guangdong, China*
[2] *ÉTS Montréal, QC, Canada*
[3] *School of Mathematics and Statistics, Xidian University, Xi'an, Shanxi, China*

## Abstract

A Laplacian pyramid-based complex neural network, CLP-Net, is proposed to reconstruct high-quality magnetic resonance images from undersampled k-space data. Specifically, three major contributions have been made: 1) A new framework has been proposed to explore the encouraging multi-scale properties of Laplacian pyramid decomposition; 2) A cascaded multi-scale network architecture with complex convolutions has been designed under the proposed framework; 3) Experimental validations on an open source dataset fastMRI demonstrate the encouraging properties of the proposed method in preserving image edges and fine textures.

**Keywords:** Deep learning, complex convolution, Laplacian pyramid decomposition

## 1. INTRODUCTION

Fast magnetic resonance imaging (MRI) is vital for clinical applications. Accordingly, different MR acceleration approaches have been proposed, such as designing physics-based fast imaging sequences (Oppelt et al., 1986), improving hardware-based parallel imaging techniques (Lustig and Pauly, 2010), and developing signal processing-based image reconstruction from undersampling k-space strategies (Lustig et al., 2008; Wang et al., 2018). Among them, k-space undersampling is one most popular method that can achieve near to perfect high-resolution images under very high acceleration factors with elaborately designed reconstruction algorithms.

Recently, deep learning has found wide employment in the field of medical imaging. It has been involved in different aspects of medical image analysis, including image reconstruction (Wang et al., 2019; Chen et al., 2019; Wang et al., 2020), segmentation (Yang

---

[*] Contributed equally

[†] Corresponding author

et al., 2019; Qi et al., 2019), and classification (Zhou et al., 2018). Deep learning was first introduced to MR image reconstruction by Wang et al., where a Convolutional Neural Network (CNN) was used to learn the nonlinear mapping between images reconstructed from the undersamled and fully sampled k-space (Wang et al., 2016). Subsequently, many deep learning-based reconstruction algorithms have been developed. Schlemper et al. proposed a deep cascaded CNN (DCCN) for dynamic MR image reconstruction (Schlemper et al., 2017). Generative Adversarial Network (GAN)-based reconstruction methods were utilized to alleviate the blurry issue identified in reconstructed MR images (Shende et al., 2019; Quan et al., 2018) . Hammernik et al. developed a Variational Network (VN)-based method to improve the image reconstruction quality (Hammernik et al., 2018). Sun et al. designed an ADMM-Net based on the classic Alternating Direction Method of Multipliers (ADMM) algorithms to obtain accurate reconstruction results from undersampled k-space data (Sun et al., 2016). A Model-based Deep Learned priors framework (MoDL) was proposed to combine the power of data-driven deep learning with model-based reconstruction schemes (Aggarwal et al., 2018).

Despite all the successes achieved, reconstruction results of existing methods are still plagued by the notorious blurry issue of tissue and the missing details, especially when high acceleration factors are utilized. To this end, we propose a novel framework with Laplacian pyramid-based complex neural network (CLP-Net) to further utilize the prior information of the available big datasets. This framework explores the multi-scale properties of Laplacian pyramid decomposition for MR reconstruction from undersampled k-space data. A cascaded multi-scale network architecture with complex convolutions is embedded in the proposed framework, as well as a data consistency layer. Experimental results on the fastMRI dataset show that our method obtains better reconstruction results than the state-of-the-art.

## 2. METHODS

### 2.1. Problem Formulation

In MR reconstruction, $y \in \mathcal{Y}$ is the noisy measurement and $x \in \mathcal{X}$ denotes the image to be recovered. That is:

$$y = \mathcal{T}x + n \tag{1}$$

where $\mathcal{T} = PF$ is an undersampled Fourier encoding matrix, and $n$ is the measurement noise modelled as additive white Gaussian noise. The corresponding fully sampled k-space data can be represented as $Fx$ and $F$ indicates the Fourier encoding matrix. $P$ denotes the undersampling mask selecting lines in k-space to be sampled. The Fourier encoding matrix is normalized as $F^H F = I$. $H$ is $Hermitian$ transpose operation.

The task of this paper is to reconstruct high-quality MR images from undersampled k-space data, whose objective function can be written as:

$$\underset{\Theta}{\operatorname{argmin}} \|x - f_{\mathrm{CNN}}(x_u; \theta)\|^2 \tag{2}$$

where $f_{\mathrm{cnn}}$ denotes the proposed reconstruction network which takes the zero-filled image $x_u$ as inputs and outputs the corrsponding high-quality images.

## 2.2. Complex Laplacian Pyramid Decomposition

Laplacian pyramid is an image encoding method (Burt and Adelson, 1983), which treats operators in different scales as its basic functions. It has a low level of computational complexity and is easy to implement. In this paper, we modify the conventional Laplacian pyramid method to make it suitable for complex-valued images. Specifically, $g_0$ is the image to be decomposed with a shape of $h \times w \times 2$ (2 means concatenating the real part and the imaginary part). $g_1$ is obtained with $g_0$ passing through a Gaussian low-pass filter. Subtracting $g_1$ from $g_0$, we can get $L_0$, which indicates the difference between $g_0$ and $g_1$:

$$L_0 = g_0 - g_1 \tag{3}$$

Afterward, $g_1$ is downsampled. By repeating the above steps with the downsampled images, a complex Laplacian pyramid $\{L_0, L_1, L_2, \dots L_n\}$ can be finally obtained. $g_0$ can be reconstructed from $g_1$ and $L_0$, which means we only need to handle the reconstruction of $g_1$ instead of $g_0$. As a result, the computation can be reduced as $g_1$ has a smaller image matrix than $g_0$.

## 2.3. Complex Convolution

MR k-space data are complex-valued data. Accordingly, the reconstructed images are also complex-valued. Although the amplitude image is frequently utilized as a simplification of the complex-value MR images, the phase image can also provide valuable information. Therefore, building a complex convolution operation-based neural network is a more effective way for MR image reconstruction.

Complex convolutions were proposed in (Wang et al., 2020). The complex-valued image $x$ can be denoted by $x_{\text{real}}$ and $x_{\text{imag}}$ and $x = x_{\text{real}} + i x_{\text{imag}}$. We can use a complex convolution filter $\Omega = \Omega_{\text{real}} + i\Omega_{\text{imag}}$ to convolve it. The components of $\Omega$, $\Omega_{\text{real}}$ and $\Omega_{\text{imag}}$, are real-valued. The convolution operation is shown in Equation (4). Since convolution operation is distributive. $\Omega$ can be written in the rectangular form as $\Omega = |\Omega|e^{i\theta} = \Omega_{\text{real}} + i\Omega_{\text{inag}} = |\Omega| \cos \theta + i|\Omega| \sin \theta$, where $\theta$ and $|\Omega|$ are the phase and magnitude of $\Omega$, respectively. In order to reduce the risk of gradient vanishing, we use Rayleigh distribution to initialize the magnitude of $\Omega$ and use the uniform distribution between $-\pi$ and $\pi$ to initialize the phase of $\Omega$.

$$\Omega * x = (\Omega_{\text{real}} * x_{\text{real}} - \Omega_{\text{imag}} * x_{\text{imag}}) + i (\Omega_{\text{real}} * x_{\text{imag}} + \Omega_{\text{imag}} * x_{\text{real}}) \tag{4}$$

## 2.4. Network Structure

Directly train a deep learning network with image pairs obtained from undersampled and fully-sampled k-space data is computational intensive. To solve this problem, we improve the conventional Laplacian pyramid to make it suitable for processing complex-valued images and built CLP-Net based on it. The architecture of Laplacian pyramid-based block in CLP-Net is shown in Figure 1.

Predictions (zero-filled image for the first layer) from the previous layers are fed into two network branches, one of which performs the complex Laplacian pyramid decomposition, and the other implements shuffle downsampling. Complex Laplacian pyramid decomposition produces two Laplacian error maps in different scales and one Gaussian map. The

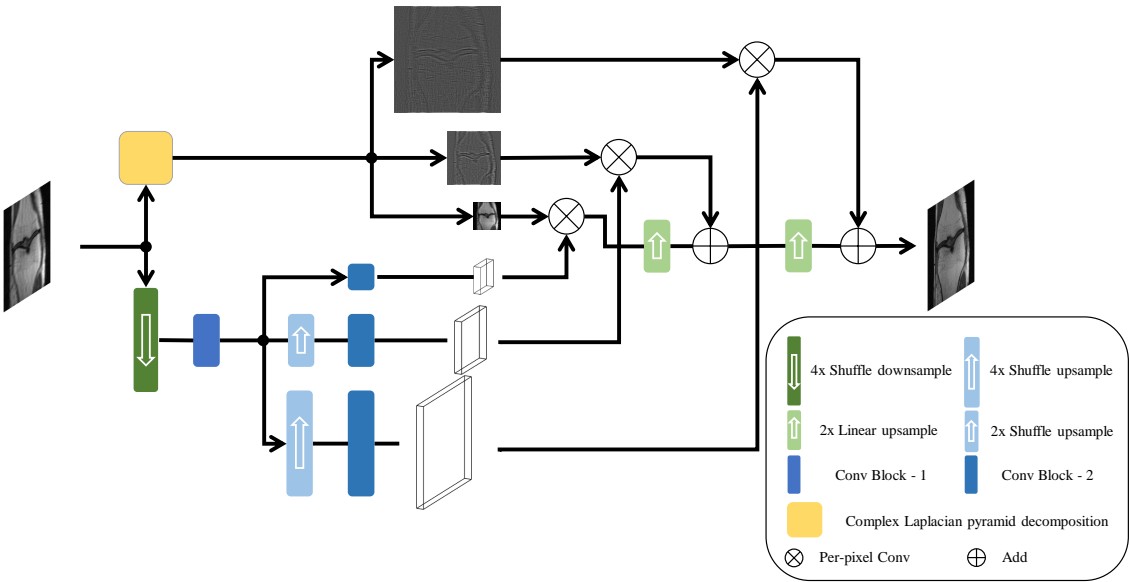

Figure 1: Laplacian pyramid-based block in CLP-Net.

shuffle downsampling module is illustrated in Figure 2. After downsampling, the resultant feature maps will be sent to the Conv Block-1 module shown in Figure 2 to extract shallow features. Conv Block-1 module consists of sixteen residual blocks, each consisting of two complex convolution layers and one ReLU activation layer. The extracted shallow features are then sent to the next three branches. Each branch has a Conv Block-2 module and a shuffle upsampling module except for the first branch, as illustrated in Figure 2. Conv Block-2 module consists of three complex convolution layers and three ReLU activation layers. Then we reshape the last dimension of the resultant deep features from Conv Block -2 module to $k \times k$ and we call the reshaped features "Kernel". Besides, k is manually set. Then, we perform the following operations

$$F(m, n) = I_{neighbor}(m, n) \otimes K(m, n) \tag{5}$$

where $K(m, n)$ means the pixel value of the "Kernel" at position $(m, n)$. And $I_{neighbor}(m, n)$ is the neighborhood of the pixel of the Laplacian error maps or the Gaussian maps at position $(m, n)$. In this paper, we investigate the effect of different k values on the reconstruction results and choose the optimal k. $F(m, n)$ is the pixel of the acquired feature maps $F$ at position $(m, n)$, and $\otimes$ means inner product operation.

After the above operations, we can obtain the resultant image by performing Laplacian reconstruction and upsampling on the acquired feature maps $F$. The resultant image will be passed trough a data consistency (dc) layer which is first proposed by (Schlemper et al., 2017). And the above whole process will be repeated several times, resulting into a cascaded structure.

(CLP-NET)

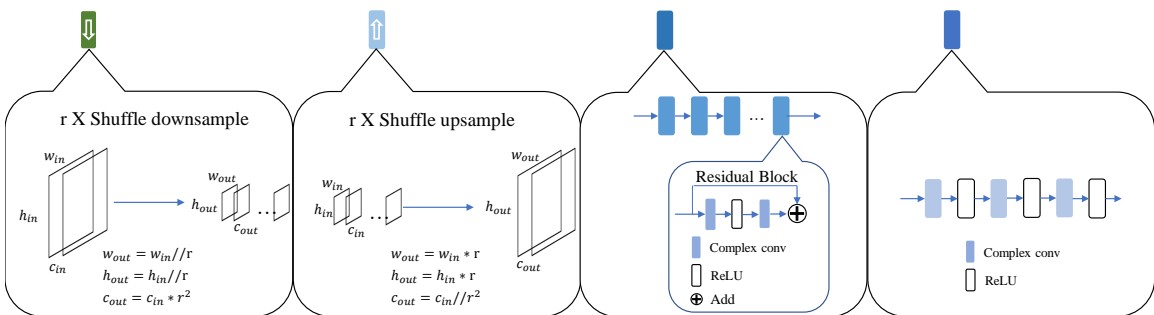

Figure 2: Illustration of the shuffle downsampling modules.

## 3. EXPERIMENTS

### 3.1. Experimental Data and configuration

In this paper, we evaluated our method on the public dataset fastMRI (Zbontar et al., 2018). The training, validation and testing subsets of the single-coil k-space data include 973, 199 and 118 scans respectively. Each scan is collected with one of of the three clinical 3T systems (Siemens Magnetom Skyra, Prisma and Biograph mMR) or one clinical 1.5T system (Siemens Magnetom Aera). Conventional Cartesian 2D Turbo Spin-Echo (TSE) protocol was used to acquire the dataset. Sequence parameters are, as per standard clinical protocol, matched as closely as possible between the systems, with a repetition time (TR) ranging between 2200 and 3000 milliseconds, and echo time (TE) between 27 and 34 milliseconds. The size of images scanned by different systems keep the same with a matrix size of $320 \times 320$, in-plane resolution of 0.5mm $\times$ 0.5mm, slice thickness of 3mm, and there is no gap between adjacent slices. We used 1D random sampling pattern with the acceleration factor of 4 and 8 auto-calibration (ACS) lines for all the experiments.

### 3.2. Implementation Details

The model was implemented with Pytorch 1.4 and trained with a NVIDIA GeForce RTX 2080Ti GPU. We used Adam algorithm (Kingma and Ba, 2014) to optimize the proposed network. The hyperparameters of Adam algorithm were set as $\beta_1 = 0.9$ and $\beta_2 = 0.999$. The loss function of proposed network is L1 loss. The number of epochs is 200. The initial learning rate was set to $\alpha = 1.0 \times 10^{-4}$ and the exponential decay rates of the learning rate for per-epoch is 0.95.

### 3.3. Quantitative Evaluation Index

Peak Signal to Noise Ratio (PSNR) and Structural Similarity Index Matrix (SSIM) are chosen to evaluate the quality of the reconstructed images. Theoretically, images with higher PSNR and higher SSIM have better quality.

## 3.4. Ablation Study

### 3.4.1. Ablation Study for the Kernel Size

Table 1: The average value of PSNR and SSIM of the reconstructed images and floating-point operations per second (GFLOPs) per epoch of CLP-net with different kernel sizes.

|       | SSIM  | PSNR  | GFLOPs |
|-------|-------|-------|--------|
| k=5   | 0.633 | 28.19 | 196    |
| k=9   | 0.634 | 28.21 | 283    |
| k=11  | 0.635 | 28.26 | 345    |
| k=13  | 0.637 | 28.29 | 419    |

Table 2: The average value of PSNR and SSIM of the reconstructed images and floating-point operations per second (GFLOPs) per epoch of CLP-net with different cascaded structure.

| cascade | SSIM  | PSNR  | GFLOPs |
|---------|-------|-------|--------|
| 3       | 0.624 | 28.02 | 117    |
| 4       | 0.628 | 28.13 | 156    |
| 5       | 0.633 | 28.19 | 196    |
| 6       | 0.635 | 28.22 | 235    |

We utilized different-sized convolution kernels to test the influence on reconstruction results. As shown in Table 1, although the network performance improves with the increase of kernel size, the improvement is marginal. However, an increase in the size of the convolution kernel leads to a sharp increase in the number of network parameters. Considering the computational efficiency and network performance, we selected a convolution kernel with the size of 5 in our proposed CLP-net.

### 3.4.2. Ablation Study for the Cascaded Structure

We implemented the network with different number of cascading modules to test the influence on network capacity. As shown in Table 2, the reconstruction performance improves with the increase of the number of cascading modules. It proves that the cascaded structure is more applicable to MR image reconstruction. To balance the network performance with the computational complexity, we utilized 5 cascading modules in our proposed CLP-net.

### 3.4.3. Ablation Study for the Laplacian pyramid decompose

In order to verify the contribution from the Laplacian pyramid decomposition to MRI image reconstruction, we compared the reconstruction results of CLP-net with and without the

Laplacian pyramid decomposition. The mean values of PSNR and SSIM are shown in Table 3. It can be observed Laplacian pyramid decomposition improves the reconstruction performance by capturing multi-scale image information.

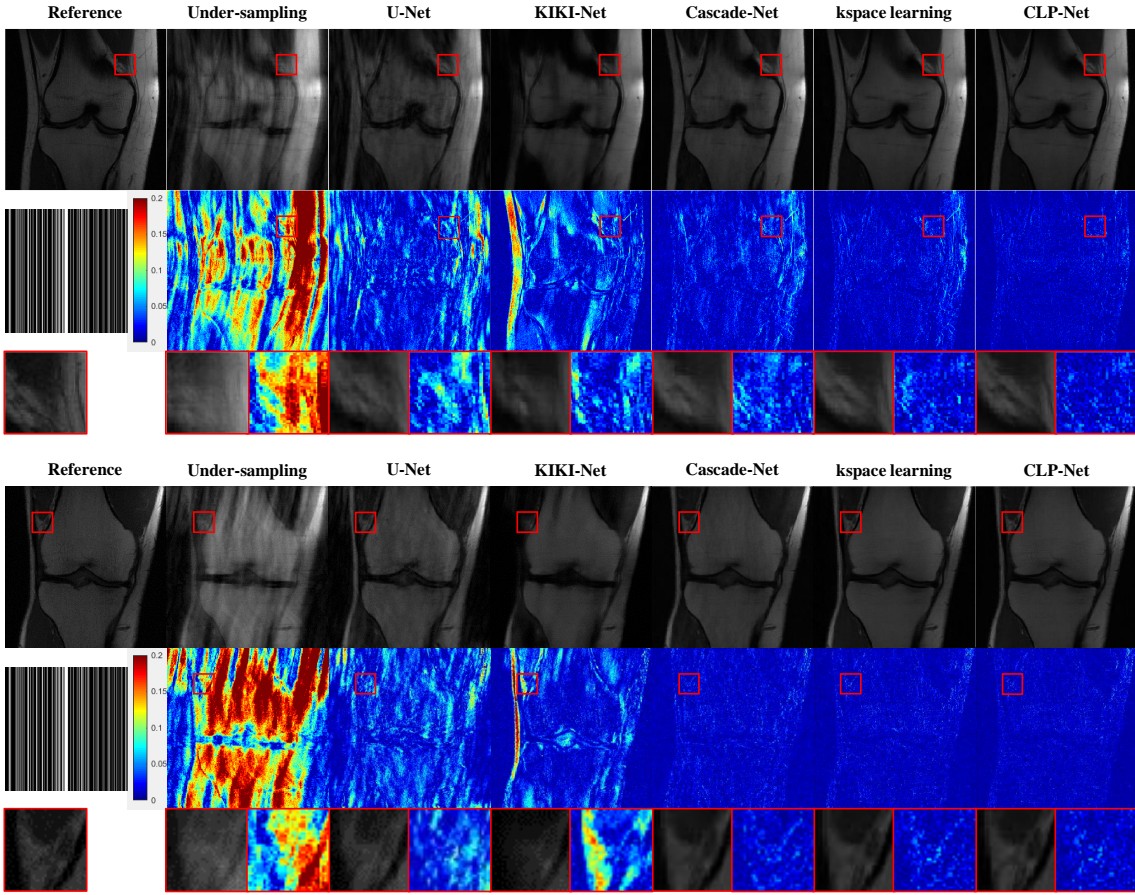

Figure 3: Reconstruction results on fastMRI dataset included the original image and the error map image of different networks. From left to right: reference; zero-filled; U-net; KIKI-net; Cascade-net; k-space learning and CLP-net.

### 3.4.4. ABLATION STUDY FOR THE COMPLEX CONVOLUTION

The average value of PSNR and SSIM of the reconstruction results of CLP-net with complex-valued convolution and the CLP-net with real valued convolution is shown in Table 4. As we can see, complex-valued convolution improves the reconstruction performance of CLP-net. Since the MRI image itself is a complex-valued image, complex convolution is more suitable for MRI reconstruction than real-valued convolution.

Table 3: The average value of PSNR and SSIM of the reconstructed images processed by CLP-net with and without Laplacian pyramid decompose.

|  | SSIM | PSNR |
|---|---|---|
| LP decompose | 0.633 | 28.19 |
| no decompose | 0.613 | 27.33 |

Table 4: The average value of PSNR and SSIM of the reconstructed images processed by CLP-net with complex convolution and CLP-net with common convolution.

|  | SSIM | PSNR |
|---|---|---|
| Complex conv | 0.633 | 28.19 |
| Common conv | 0.624 | 27.96 |

## 3.5. Comparing with Other Methods

Table 5: The average value of PSNR and SSIM of the reconstruction images processed by different networks on fastMRI dataset.

|  | Zero-filled | U-net | KIKI-net | Cascade-net | kspace learning | CLP-net |
|---|---|---|---|---|---|---|
| **PSNR** | 23.23 | 25.28 | 25.95 | 27.74 | 27.78 | **28.19** |
| **SSIM** | 0.518 | 0.555 | 0.583 | 0.618 | 0.622 | **0.633** |

To evaluate the reconstruction performance of the proposed network, we also trained U-net (Ronneberger et al., 2015), KIKI-net (Eo et al., 2018), Cascade-net (Schlemper et al., 2017) and k-space learning (Han et al., 2019) for the comparison purposes. The reconstruction results are shown in Figure 3.

As we can see in Figure 3, all the methods are able to generate high-quality images. The suppression of artifacts improves the visual quality of the reconstruction images greatly. Nevertheless, there are differences between the reconstruction results of different networks. The reconstruction effect of U-net is relatively poor compared to other networks (both the inputs and outputs of U-net are the respective amplitude MR images). There are slight artifacts in the reconstructed images of U-net shown in Figure 3. In the zoomed regions-of-interest (ROI) marked by the red rectangle, the over-smoothing phenomenon can be observed in the reconstruction results of U-net. It leads to the loss of many texture details. Besides, there are more reconstruction residuals in the results of U-net than that of the other networks as shown in the error-map (Figure 3). Although noise suppression ability of KIKI-net is better than U-net, the error-map shows that high-intensity noise still exists in some areas. As we can see in the zoomed ROI in Figure 3, KIKI-net successfully suppressed the artifacts but lost many texture details. Cascade-net not only has better noise

suppression capability than U-net and KIKI-net but also has excellent artifact suppression capability. As shown in Figure 3, the suppression of artifacts improves the visual quality of the reconstructed image of Cascade-net significantly. The over-smoothing phenomenon in the reconstruction results of Cascade-net is relatively light. However, the texture detail loss is still a serious problem shown in the zoomed ROI (Figure 3). Based on the Cascade-net reconstruction results, k-space learning further suppresses the reconstruction noise. The reconstructed images of k-space learning shown in Figure 3 are closer to the reference images visually than that of U-net, KIKI-net, and Cascade-net. More texture information was saved in the reconstructed image of k-space learning. However, blurry effects can still be observed. The reconstruction performance of CLP-net is the best among the different methods. As we can see in Figure 3, reconstructed images of CLP-net possess more texture details and less image noises, which proves that CLP-net has an excellent reconstruction ability. Compared with the reconstruction results of the other networks, the tissue in the reconstructed images of CLP-net is clearer shown in the zoomed ROI in Figure 3. It also proves that there is no excessive noise reduction in the reconstruction process of CLP-net.

We calculated the average PSNR and SSIM of the whole test dataset. The results are shown in Table Table 5. The proposed CLP-net achieves the best scores for both PSNR and SSIM.

## 4. CONCLUSIONS

In this paper, we propose a novel Laplacian pyramid-based complex neural network for fast MR imaging. Complex Laplacian pyramid decomposition provides encouraging multi-scale properties for MR reconstruction from undersampled k-space data. The proposed framework contains a cascaded multi-scale network architecture with complex convolutions and data consistency layers. Experimental results demonstrated that our method achieved comparable and even superior reconstruction results than recently published state-of-the-art methods both quantitatively and qualitatively.

## Acknowledgments

This work was supported by the National Natural Science Foundation of China (61871371, 81830056, 61671441), Science and Technology Planning Project of Guangdong Province (2017B020227012, 2018B010109009), the Basic Research Program of Shenzhen (JCYJ201805 07182400762) and Youth Innovation Promotion Association Program of Chinese Academy of Sciences (2019351).

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
