# OpenReview forum: "Laplacian pyramid-based complex neural network learning for fast MR imaging"
_MIDL.io/2020/Conference — MIDL 2020_

### Official Review · AnonReviewer2 · 2020-03-08
**Review for "Laplacian pyramid based complex neural network learning for fast MR imaging"**

**Rating:** 3
**Confidence:** 4
**Recommendation:** Poster

**Summary:**

This paper proposes to learn Laplacian pyramid based complex neural network (CLP-Net) for high-quality image reconstruction from undersampled k-space data. The goal is to accelerate MR imaging. The experimental results on in vivo datasets show that the proposed method obtains better reconstruction performance than three state-of-the-art methods.

**Strengths:**

1) a new framework for MR reconstruction from undersampled k-space data has been proposed by exploring the encouraging multi-scale properties of Laplacian pyramid decomposition;

2) a cascaded multiscale network architecture with complex convolution has been designed under the proposed framewor

3) The experimental validations on in vivo datasets have shown higher potential of this method in preserving the edges and fine textures when compared to other state-of-the-art methods.

**Weaknesses:**

No notable weakness identified.
I support this paper due to the novel cascaded multiscale network architecture using complex convolutions, and its strong performance on in vivo datasets in preserving the edges and fine textures.

**Justification Of Rating:**

See above. I support this paper due to the novel cascaded multiscale network architecture using complex convolutions, and its strong performance on in vivo datasets in preserving the edges and fine textures.

**Paper Type:**

both

**Special Issue:**

yes

---

> ### Author Response · Authors · 2020-03-31
> **Thanks for the recognition and our code will be released**
>
> Thanks for the recognition. We will release our code after it gets accepted.

---

### Official Review · AnonReviewer3 · 2020-03-14
**Office review**

**Rating:** 2
**Confidence:** 4

**Summary:**

This paper proposes a Laplacian pyramid based complex neural network for fast MRI imaging. The proposed deep networks contains two important component: Laplacian pyramid and the complex convolution, which are existing work. The authors compares the proposed network with several existing methods and shows its better performance.

**Strengths:**

The paper tries to improve MRI reconstruction with Laplacian pyramid and complex convolution. The paper compares the proposed methods with several existing methods. The paper is well organized and presented.

**Weaknesses:**

1. The paper fails to cover a series of recent work on MRI reconstruction with Generative Adversarial Networks including:
*Shende, Priyanka, Mahesh Pawar, and Sandeep Kakde. "A Brief Review on: MRI Images Reconstruction using GAN." 2019 International Conference on Communication and Signal Processing (ICCSP). IEEE, 2019.
*Quan, Tran Minh, Thanh Nguyen-Duc, and Won-Ki Jeong. "Compressed sensing MRI reconstruction using a generative adversarial network with a cyclic loss." IEEE transactions on medical imaging 37.6 (2018): 1488-1497.

I believe GAN-based MRI reconstruction could alleviate the blurry issue in reconstruction, but the authors have not included any reference, discussion or comparison with such methods.

2. The authors do not give any ablation study of the proposed model. Why combining Laplacian pyramid and the complex convolution could improve the performance? Which of these two components play an important role and whether both components improve the performance?


**Justification Of Rating:**

The paper fails to mention a series of related research with GAN on MRI imaging. No ablation study is given to analyze the proposed model. Then I could not give any rating beyond weak reject, unless the authors improve the paper.

**Paper Type:**

methodological development

**Special Issue:**

no

---

> ### Author Response · Authors · 2020-03-27
> **The ablation study  and the citations on GAN based MR imaging have been supplemented.**
>
> Thank you for your comments.
> 1. Following your advices, we have added the citations on GAN based MR imaging.
> 2. Regarding your question why laplacian pyramid and the complex convolution could improve the performance？ We have supplemented the ablation study to investigate the two components' contributions in the final reconstruction.
>
> #                                           #SSIM     #PSNR
> Laplacian pyramid            #0.631     #28.16
> no Laplacian pyramid      #0.613     #27.33
>
> #                                           #SSIM     #PSNR
> complex convolution        #0.633     #28.19
> normal convolution          #0.624     #27.96

---

### Official Review · AnonReviewer5 · 2020-03-19
**Interesting approach but experiments lack conviction**

**Rating:** 2
**Confidence:** 3

**Summary:**

The study presents a new framework for zero-filled MRI reconstruction. The framework is built on a two-way backbone, simultaneously processing from its Laplacian pyramid decomposition and a downsampled version of the input signal. It is built in an end-to-end deep learning model involving complex convolutions. Experiments are performed on a in-house dataset with 2D images from 22 patients, on which the proposed approach is reported with the best performances compared to state-of-the-art community approaches.

**Strengths:**

Substantial effort is done to formulate the studied problem. There are great visual supports including deep learning architecture pipeline and sample results. Authors also make efforts to explain the theorical bases of the components used in their approach, such as complex convolutions or complex Laplacian pyramid decomposition. Experiments report conventional measures for this task.

**Weaknesses:**

From an evaluation perspective, all results are derived from an in-house dataset which makes the paper unreproducible. Most importantly, there does not seem to be any validation set, while the paper proposes a new architecture (i.e. an optimization hyper-parameter). The paper suggests that experiments are on 2D images, while competing approaches such as KIKI-Net report experiments on 3D images. If experiments are on 2D, authors should specify whether the training/testing split was done patient-wise or if images from a patient can be in both sets.

**Detailed Comments:**

- figures with visual examples could be enhanced with zoomed-in region with results for all benchmarked methods; also, both figures with samples could be merged
- acronym "dc layer" is never introduced
- 2.3, when introducing the notation with cos and sin, "where Theta and Omega" should be "|Omega|"
- it is not clear how the unet is implemented with complex inputs

**Justification Of Rating:**

The major drawback of the paper is the experiment evidence from the designed setup: experiments are in 2D and on in-house data including few patients, with no validation set. Although state-of-the-art methods are reimplemented, there are no direct comparison on publicly available data.

**Paper Type:**

both

**Questions To Address In The Rebuttal:**

- How have the training set and testing sets been used ? The split between training/testing should be explicited, along with total number of images.
- There are public datasets suited for the task (cf other community works), which would produce insightful measures comparable to other reported performance and reproducible. Would it be possible to perform experiments on such public dataset, or maybe report generalization performance of the proposed approach?

**Special Issue:**

no

---

> ### Author Response · Authors · 2020-03-27
> **Details and our code will be supplemented and new experimental results on the public dataset have been given**
>
> Thank you for your comments.
> 1. We will supplement the data splitting details and release our code in Github. All our data were splitting in patient wise.  We use the training set of the open source dataset fastMRI to train our network, the test set for validation, the validation set for evaluating. The details of the fastMRI dataset have been added.
> 2. New results on the public dataset fastMRI have been supplemented to enhance the reproducibility.
>
> #index #Zero-filled #U-net #KIKI-net #Cascade-net #kspace learning #CLP-net
> #PSNR #23.23          #25.28  #25.95      #27.74             #27.78                    #28.19
> #SSIM  #0.518         #0.555   #0.583     #0.618             #0.622                     #0.633
>
> 3. We have enhanced the results with zoomed-in region for the visual comparison.
> 4. We have provided the explantions for the DC layer, namely data consistency layer. We also have supplemented a citation for others to refer to for more details.
> 5. The typo on the $|Omega|$ have been revised.
> 6. The inputs and outputs are both magnitude for the U-net. We have added this in the revised manuscript.

---

### Official Review · AnonReviewer6 · 2020-03-19
**Lacks critical detail about methodology**

**Rating:** 2
**Confidence:** 3

**Summary:**

The paper proposes a laplacian pyramid-based CNN for reconstruction of MR images from undersampled k-space data to accelerate MRI acquisition. The authors have demonstrated using a Laplacian pyramid-based scheme to recover undersampled k-space data and reconstruct MR images. Results with other state-of-the-art methods show an improvement in PSNR and SSIM on a brain MRI dataset.

**Strengths:**

An approach to reconstruction undersampled MR images and accelerate MR imaging, which results in higher PSNR and SSIMs on a brain MRI dataset, compared to other state-of-the-art approaches like U-Net, Cascade-Net, and PD-Net.

**Weaknesses:**

The paper lacks critical details on the network architecture---the loss function used, the architecture of convolutional layers, a well-structured and well-formed figure representation the network, the cascaded structure of the proposed architecture, the datasets used---to name a few, as well as an ablation study, both on the width of the CNN as well as the cascaded architecture. While the results do indeed beat state-of-the-art, I believe it is not straightforward to reproduce the results from the manuscript in its current form.
The pipeline also includes an inverse Fourier transform and it is not clear whether the entire network is trained with backpropagation, and if so, how.

**Justification Of Rating:**

While the results do beat state-of-the-art, I believe the manuscript can be accepted after some critical revision in terms of description of methology and datasets. In my opinion, reproducibility of a paper strengthens the results of the paper.

**Paper Type:**

validation/application paper

**Questions To Address In The Rebuttal:**

A more rigorous description of the overall method, with proper descriptions of the model and loss functions is imperative. A more detailed description of the dataset is also necessary, as the methodology figure seems to use knee MRI, but results are demonstrated on brain MRI.

**Special Issue:**

no

---

> ### Author Response · Authors · 2020-03-27
> **Details will be supplemented and the code will be released in the Github**
>
> Many thanks for your comments. To address your concerns, we have made the following changes.
> 1. Following the advices, we will supplement the details regarding the dataset, network architecture and release our code.  We also will display the results on the knee dataset as well. Thanks.
> （1）We have added a more detailed figure of  the cascade structure, the shuffle downsample and shuffle upsample, and the convolution blocks;
> （2）We redrew the figure of the network architecture;
> （3）We use the L1 loss as our loss function;
> （4）We present the results on the open source dataset fastMRI (knee) to enhance the reproducibility of our results;
>
> #index #Zero-filled #U-net #KIKI-net #Cascade-net #kspace learning #CLP-net
> #PSNR #23.23          #25.28  #25.95      #27.74             #27.78                    #28.19
> #SSIM  #0.518         #0.555   #0.583     #0.618             #0.622                     #0.633
>
> （5）The inverse Fourier transform is embedded in the dc layer (data consistency layer), and we have added the details of the dc layer；
>
> 2. Regarding the Fourier transform, it is a bulit-in function in pytorch. We indeed used backpropagation with autograd package in pytorch for the network training. The code for both training and testing will be released.
>
> 3. We have added the results of the ablation study on the kernel size and the cascade structure.
>
> #kernel size   #SSIM   #PSNR   #GFLOPS
> #k=5               #0.633   #28.19    #196
> #k=9               #0.634   #28.21    #283
> #k=11             #0.635   #28.26    #345
> #k=13             #0.637   #28.29    #419
>
> #cascade        #SSIM    #PSNR    #GFLOPS
> #cascade=3    #0.624    #28.02    #117
> #cadcade=4    #0.628   #28.13     #156
> #cascade=5     #0.633   #28.19     #196
> #cascade=6     #0.635   #28.22     #235

---

> > ### Comment · AnonReviewer6 · 2020-04-03
> > **Needs more details**
> >
> > I thank the authors for their response. While concerns regarding datasets and ablation study have been addressed, I regret that concrete information on the pipeline, loss function, convolutional models still hasn't been provided in the rebuttal.
> >
> > * Details on the shuffle down- and up-sampling could be provided here instead of the revised manuscript.
> > * The loss function is L1, but how is it computed? What are predictions and the targets? If it is the reconstruction loss, was anything done to account for smooth and unrealistic results generated by L1 and L2 reconstruction losses? Is the loss applied at only the final output, or at the outputs of each block in the cascade?
> >
> > There are several details that are still unclear:
> > * In equation (6), I_neighbour(m,n) is the neighbourhood around pixel (m,n), but pixel (m,n) in which image?
> > * In the last paragraph on Page 6, the result of the convolutional operations is passed through an inverse Fourier transform layer, followed by a data consistency  layer. What the data consistency layer is, is still not clear.
> > * Continuing with the previous concern, the conversion between image and frequency domains seem to be arbitrary. There is an inverse Fourier transform operation after each block in the cascade, but it is not clear where the conversion to frequency domain happens.
> >
> > Furthermore, several modules in the network haven't been motivated enough. For example, why do we expect a cascaded architecture to imporove the result in this Laplacian pyramid decomposition scheme?
> >
> > Finally, the language needs to be modified substantially.
> >
> > Based on these concerns, I am not inclined to change my final rating.

---

> > > ### Author Response · Authors · 2020-04-04
> > > **We have supplemented more details.**
> > >
> > > Many thanks for your comments.
> > >
> > > 1. Shuffle-down and Shuffle-up were proposed in the paper "Real-Time Single Image and Video Super-Resolution Using an Efficient Sub-Pixel Convolutional Neural Network" by Shi et. al (2016).  Shuffle-up rearranges elements in a tensor of shape (Batch, Channel*r*r, Height, Weight) to a tensor of shape (Batch, Channel, Height*r, Weight*r), and r is the upscale factor.  Shuffle-down rearranges elements in a tensor of shape (Batch, Channel, Height*r, Width*r) to a tensor of shape (Batch, Channel*r*r, Height, Width), and r is the downscale factor.  And our pseudocode is here:
> > >
> > > def ComplexShuffleDown(inputs, factor):
> > >         batch, channel_in, height_in, width_in, complex_channel = inputs.size()
> > >         channel_out = channel_in * factor** 2
> > >         height_out = height_in // factor
> > >         width_out = width_in // factor
> > >         output = inputs.view(batch, channel_in, height_out , factor, width_out, factor, complex_channel)
> > >         output = output.permute(0, 1, 5, 3, 2, 4, 6)
> > >         output = output.view(batch, channel_out , height_out , width_out, complex_channel)
> > >         return output
> > >
> > > def ComplexShuffleUp(inputs, factor):
> > >         batch, channel_in, height_in, width_in, complex_channel = inputs.size()
> > >         channel_out = channel_in // (factor ** 2)
> > >         height_out = height_in * factor
> > >         width_out = width_in * factor
> > >         output = x.view(batch, channel_out, factro, factor, height_in , width_in, complex_channel)
> > >         output = output.permute(0, 1, 4, 3, 5, 2, 6)
> > >         output = output.view(batch, channel_out , height_out , width_out, complex_channel)
> > >         return output
> > >
> > > 2.  We didn't use L2 loss. The L1 loss we used is a built-in function in Pytorch, so we don't need to compute it ourself. The L1 loss is applied at the outputs of each block in the cascade, that means the predictions and the targets are the reconstructed MR images in each cascade and the fully sampled MR images, respectively.
> > >
> > > 3. The reconstruction performance of CLP-net is the best in this paper. As
> > > we can see in Figures 4, more texture details, and less noise prove that CLP-net has excellent
> > > reconstruction ability. Compared with the reconstruction results of the other networks, the
> > > tissue in that of CLP-net is a clearer reference to the zoomed ROI in Figures 4. It proves
> > > that there is no excessive noise reduction in the reconstruction process of CLP-net. CLP-net improves on the disadvantage of being too smooth and provides more realistic results.
> > >
> > > 4. From the CLP-Net architecture, there are two Laplacian error maps and one Gaussian map, they have different sizes. Each of them will be performed the per-pixel convolution. So, in equation (6), the pixel(m,n) are in the above maps.
> > >
> > > 5. The inverse Fourier transform actually is a part of the dc layer (data consistency layer) which was proposed in the paper "A Deep Cascade of Convolutional Neural Networks for Dynamic MR Image Reconstruction". The whole pipeline consist of several (we set it to 5) cascaded structures, each cascaded structure consist of a CLP-Net and a dc layer.  And we will put the undersampled MR images into the CLP-Net and it will output the reconstructed MR images. Then the reconstructed MR images will be put into the dc layer with the mask and the undersampled k-space data. More specifically, first, the input reconstructed MR images will be performed a Fourier Transform and multiplied by a inverse mask afterwards. The inverse mask is obtained by (1 - mask). After that, we can obtain the reconstructed k-space data by adding the undersampled k-space data on the results of the previous step, and we will perform inverse Fourier Transform on the reconstructed k-space data to produce the final reconstructed MR images.  The dc layer can preserve more real details.
> > >
> > > 6.
> > > #                                           #SSIM     #PSNR
> > > Laplacian pyramid            #0.633     #28.19
> > > no Laplacian pyramid      #0.613     #27.33
> > >
> > > #cascade        #SSIM    #PSNR    #GFLOPS
> > > #cascade=3    #0.624    #28.02    #117
> > > #cadcade=4    #0.628   #28.13     #156
> > > #cascade=5     #0.633   #28.19     #196
> > > #cascade=6     #0.635   #28.22     #235
> > >
> > > From the results of ablation study, we can see that the cascaded structure can take advantage of the Laplacian pyramid several times, the reconstructed MR images will be more precise each time.

---

> > > > ### Comment · AnonReviewer6 · 2020-04-06
> > > > **Rating change**
> > > >
> > > > Thank you for your detailed response. I think it answers several of my initial concerns. I would suggest the authors to add these explanations in some form to the manuscript to make it much clearer.
> > > > Given the responses of the authors, I would also like to change my rating to weak accept, leaning more towards borderline (as there does not seem to be a borderline rating). Results on more datasets, more time motivating the problem and the approach, as well as revising the language of the paper would have helped increase this rating.
> > > >
> > > > PS: Please clarify also the "no laplacian pyramid" version in the revised paper if accepted.

---

### Author Response · Authors · 2020-03-30
**Top level reply**

We thank all the reviewers for recognizing our contributions in the novel design of the network structures and the better reconstruction abilities of our proposed method. The main concerns all lie in the lack of details for reproducing the results.
To address this major concern, we plan to open our codes to the public in the Github.
Futhermore, new citations, figures and experimental results have been provided to facilitate the readers to better understand our methodology.

These details will be provided in the final manuscript.  Thanks.

---

### Meta-Review · Area_Chair1 · 2020-04-06
**MetaReview of Paper315 by AreaChair1**

**Rating:** 3

**Metareview:**

The authors presented a robust rebuttal addressing the main concerns of the reviewers, providing more details and explanations about the method together with experiments in a public available dataset. Even if I agree with reviewer 3 that GAN-based reconstruction needs to be discussed more in the paper as it is used a lot from the community for reconstruction problems, I think that the methodology of the paper has merit and it can be interesting for the community. However,  I also encourage the authors to incorporate all the answers to the reviewers in their final version.

**Paper Type:**

methodological development

**Special Issue:**

no

---

### Decision · Program_Chairs · 2020-04-11

Accept